# Proteomic, Transcriptomic, Mutational, and Functional Assays Reveal the Involvement of Both THF and PLP Sites at the GmSHMT08 in Resistance to Soybean Cyst Nematode

**DOI:** 10.3390/ijms231911278

**Published:** 2022-09-24

**Authors:** Naoufal Lakhssassi, Dounya Knizia, Abdelhalim El Baze, Aicha Lakhssassi, Jonas Meksem, Khalid Meksem

**Affiliations:** 1Department of Plant, Soil and Agricultural Systems, Southern Illinois University, Carbondale, IL 62901, USA; 2Faculté des Sciences et Techniques, Université Abdelmalek Essaâdi, Tanger 90000, Morocco; 3Trinity College of Arts and Sciences, Duke University, Durham, NC 27708, USA

**Keywords:** PLP, THF, SCN resistance, SHMT, soybean, mutational analysis, composite hairy root transformation

## Abstract

The serine hydroxymethyltransferase (SHMT; E.C. 2.1.2.1) is involved in the interconversion of serine/glycine and tetrahydrofolate (THF)/5,10-methylene THF, playing a key role in one-carbon metabolism, the de novo purine pathway, cellular methylation reactions, redox homeostasis maintenance, and methionine and thymidylate synthesis. *GmSHMT08* is the soybean gene underlying soybean cyst nematode (SCN) resistance at the *Rhg4* locus. GmSHMT08 protein contains four tetrahydrofolate (THF) cofactor binding sites (L129, L135, F284, N374) and six pyridoxal phosphate (PLP) cofactor binding/catalysis sites (Y59, G106, G107, H134, S190A, H218). In the current study, proteomic analysis of a data set of protein complex immunoprecipitated using GmSHMT08 antibodies under SCN infected soybean roots reveals the presence of enriched pathways that mainly use glycine/serine as a substrate (glyoxylate cycle, redox homeostasis, glycolysis, and heme biosynthesis). Root and leaf transcriptomic analysis of differentially expressed genes under SCN infection supported the proteomic data, pointing directly to the involvement of the interconversion reaction carried out by the serine hydroxymethyltransferase enzyme. Direct site mutagenesis revealed that all mutated THF and PLP sites at the GmSHMT08 resulted in increased SCN resistance. We have shown the involvement of PLP sites in SCN resistance. Specially, the effect of the two Y59 and S190 PLP sites was more drastic than the tested THF sites. This unprecedented finding will help us to identify the biological outcomes of THF and PLP residues at the GmSHMT08 and to understand SCN resistance mechanisms.

## 1. Introduction

Soybeans are the largest source of proteins and the second-largest source of oil worldwide. The production value of soybeans in the United States amounted to USD 46.06 billion in 2020 [1]. Soybean production, however, is affected by the presence of a microscopic parasitic roundworm, soybean cyst mematode (SCN), which contributes dramatically to increased yield loss in soybean crops nationwide, causing an estimated USD 1.5 billion in damage [2]. Emerging SCN populations have adapted to the resistance found in certain varieties of soybean, rendering the plant susceptible to infection. In fact, more than 95% of cultivated soybeans in the U.S. use SCN-resistant varieties based on the PI 88788 source of resistance, and 3% of varieties carry resistance from Peking. Due to SCN adaptation, a reduction in the effectiveness of resistant cultivars is taking place [3]. The shift in virulence of the pathogen resulted in 80% of fields in Midwest having SCN that can reproduce on PI 88788 [4]. Peking-type of resistance presents a sustainable alternative to breed for soybean lines with broad resistance to SCN. Cloning novel genes and understanding Peking-type resistance is essential and key to creating soybean varieties.

Peking-type reaction has been reported to be bigenic, requiring both the *rhg1* and the *Rhg4* loci [5]. The gene underlying resistance to SCN at the *Rhg4* locus, the *GmSHMT08*, has been identified and functionally characterized [2,6,7]. Additionally, copy numbers of the *Rhg4* were shown to play an essential role in broad resistance to SCN against five nematode races [8]. Although the soybean genome encodes at least 13 GmSHMT members, only the cytosolic GmSHMT08c was shown to play a role in SCN resistance, with the absence of functional redundancy by the other GmSHMT members, including the other cytosol-targeted GmSHMT05, the four nucleic-targeted GmSHMTs, the two plastidial-targeted GmSHMTs, and the five mitochondrial-targeted GmSHMTs [7].

The serine hydroxymethyltransferase (SHMT) is commonly present in plant and animal species. SHMT plays essential role in methionine synthesis, one-carbon metabolism, and the maintenance of redox homeostasis during photorespiration [9,10,11,12]. The SHMT is involved in the interconversion of serine/glycine and tetrahydrofolate (THF)/5,10-methyleneTHF through a transaldimination reaction [9]. The enzymatic co-factor THF is involved in the biosynthesis of various biologically important molecules including purine and pyrimidine nucleotides [13]. On the other hand, PLP acts as a coenzyme in all transamination reactions and in certain decarboxylation, deamination, and racemization reactions of amino acids. PLP, the active form of vitamin B6, is required for hundreds of different reactions in human metabolism, primarily for the synthesis of amino acids and amino acid metabolites and for the synthesis and/or catabolism of certain neurotransmitters and degradation pathways [14]. The SHMT enzyme is therefore essential to directing one-carbon units to the folate-mediated one-carbon metabolism that is required for nucleotide biosynthesis, methyl group biogenesis, and vitamin and amino acid metabolism during glycine biosynthesis [15]. During serine biosynthesis, SHMT plays a major role in the photorespiration metabolic reaction and is therefore essential for C3 plants. Through the glyoxylate cycle, SHMT plays a role in the maintenance of redox homeostasis, involving the gluthatione synthase, peroxidases, and other related genes. It is known that mutations at the mitochondrial *AtSHMT1* cause a photorespiratory deficiency in the plant model *Arabidopsis thaliana* [16]. Mutations in the human SHMT protein were shown to cause cancers and cardiovascular diseases [17,18,19].

SCN resistance in Forrest is derived from Peking (PI 548402) and is considered to be a promising cultivar that confers resistance to SCN that differs from SCN resistance in PI 88788. Two naturally occurring mutations, P130R and N358Y, distinguish the Forrest *GmSHMT08* allele from the susceptible soybean alleles contained in Essex and Williams 82 [2]. The GmSHMT enzyme contains several PLP and THF binding and catalysis sites that are essential to carrying out the transaldimination reaction [7]. Recently, the two Forrest-specific polymorphic substitutions (P130R and N358Y) that differ from the susceptible Essex have been reported to impact the mobility of a loop near the entrance of the (6S)-tetrahydrofolate binding site [20]. Both ligand binding and kinetic studies indicate a severe reduced affinity for folate, which dramatically impaired enzyme activity in Forrest GmSHMT08 [20].

In the current study, we performed proteomic analysis of a set of protein complexes that was immunoprecipitated using GmSHMT08 antibodies under SCN infected soybean roots. Although soybean cyst nematodes infect soybean roots, leaves play an important role by supplementing nematodes with most of the nutrients that they use to grow and complete their life cycle. In the current study, root and leaf transcriptomic analysis of differentially expressed genes under SCN infection supported the data from LC-MS. In fact, integration of proteomic and transcriptomic data pointed to the involvement of several proteins that belong mainly to pathways that use glycine/serine as a substrate/precursor. Therefore, the obtained data pointed to the involvement of the interconversion reaction carried out by the serine hydroxymethyltransferase protein. Most importantly, site-directed mutagenesis combined with composite hairy root transformations in addition to mutational analysis of the previously identified 18 EMS *Gmshmt08* Tilling mutants derived from Forrest [2,6,7] uncovered the impact of the four THF cofactor binding sites, the four PLP cofactor binding sites, and the two PLP cofactor catalysis sites at the GmSHMT08 protein on SCN resistance. This study reveals for the first time the large effect of both PLP cofactor binding and PLP cofactor catalysis sites on SCN resistance when compared to THF cofactor binding sites.

## 2. Results

### 2.1. Mass Spectrometry Identifies the Presence of Proteins That Use Serine and Glycine as Substrates/Precursors

To identify components related to the SCN resistance mechanism, we analyzed the mass spectrometry data from the immunoprecipitated protein complex that was conducted using anti-GmSHMT08 antibodies immobilized to beads in a chromatography column. Several peptides related to SCN infection were present when comparing non-infected and SCN-infected root-eluted fractions. Under non-SCN-infected conditions, only the GmSHMT08 was present in the analyzed proteomic root fractions of Forrest and Essex soybean Appendix A. For SCN-infected soybean roots, mass spectrometry analysis showed that the obtained fragmented peptides belong to 37 proteins in the resistant genotype “Forrest” only, while three proteins were identified in the susceptible genotype “Essex” only (Figure 1). Additionally, 23 proteins were common between Forrest and Essex (Figure 1). In addition to translation, growth, cell differentiation, response to stress, flower development, and carbohydrate metabolic processes that were found in both cultivars, many other additional categories (biological processes), such as cellular metabolic process, transport, biosynthetic process, and signal transduction, were contained in the resistant cultivar only (Figure 1). When comparing the biological processes that differentiate the SCN resistance reaction from susceptibility, two categories were mainly dominant in the susceptible cultivar Essex. A total of 75% of the genes that correlate with the presence of the nematode were linked to translation versus 26% in the resistance line. Surprisingly, the growth process occupied 25% in Essex versus 5% in Forrest. The presence of these two main processes in Essex is coherent with the development and growth of plant root cells in the susceptible lines to form syncytial feeding structures.

In silico analysis of the fragmented peptides obtained from the LC-MS analysis Appendix A identified 62 genes that belong to the 37 proteins that were identified in Forrest only, 3 genes that belong to the 3 proteins that were identified in Essex only, and 74 candidate genes that belong to the 23 members that were common between Essex and Forrest (Figure 2A). As expected, mass spectrometry analysis revealed the presence of the GmSHMT08, GmSNAP18, and GmPR08-BetVI proteins, which is coherent with the previously reported physically interacting GmSHMT08/GmSNAP18/GmPR08-BetVI protein complex in resistance to SCN [21]. Interestingly, we were able to identify several proteins (i.e., glycine decarboxylase (GLDC), glycine dehydrogenase (GlyDH), serine/glycine hydroxymethyltransferase (SHMT/GHMT), etc.) that belong mainly to cycles that use glycine/serine as substrate/precursor and therefore that are directly related to the interconversion reaction carried out by the serine hydroxymethyltransferase protein Appendix A.

### 2.2. Identification of Induced Gene Expression in Response to SCN Infection

Although a nematode interacts mostly during its life cycle with soybean roots, most of the nutrients that the nematode uses to grow and complete its life cycle are transported from the leaves. Therefore, to understand and identify the biological pathways that are linked to the presence of nematodes, the current study explored the expression of genes in both leaves and roots in response to SCN infections. Using an integrated approach combining the mass spectrometry data of 136 genes (identified in Essex MS and Forrest MS infected roots) and RNAseq data of 1538 DEG (SCN-infected Forrest root) and 8282 DEG (SCN-infected Forrest leaves), we identified several genes that were differentially expressed under SCN infection in soybean roots (Figure 2C). Transcripts of these identified genes were induced up to 3.9 Log2FoldChange in the SCN-infected soybean roots and up to 10.42 Log2FoldChange in the SCN-infected soybean leaves. Transcripts were downregulated up to 5.92 Log2FoldChange in the SCN-infected soybean roots and up to 11.41 Log2FoldChange in the SCN-infected soybean leaves (Figure 2D). These data reveal the importance of soybean leaves during root SCN infections in the resistant reaction. Of 136 genes identified in Forrest MS, 78 were differentially expressed under SCN infection in the resistant Forrest, including root and leave tissues. Out of the 78 genes, 56 were differentially expressed in Forrest leaves, 8 were differentially expressed in Forrest roots, and 14 were differentially expressed in both Forrest roots and leaves. Most of the identified genes belong mainly to 30 gene families, of which most were found to be related to redox hemeostasis, serine/glycine conversion, glyoxylate cycle, glycolysis, succinyl-CoA, heme biosynthesis related enzymes, cytoskeleton-related enzymes, and ATP mitochondrial related genes (Appendix A, Figure 3).

### 2.3. Correlation between the Identified Genes and the Previously Reported QTLs for SCN Resistance

Most of the identified genes were mapped to QTLs for resistance to SCN using different mapping populations. In fact, 34 genes were located within reported SCN QTLs; 21, 4, and 2 genes were located ~3Mbp, ~4–6 Mbp, and ~11Mbp away from reported SCN QTLs, respectively Appendix A). The most reported gene that mapped to QTLs for resistance to SCN is the glutamine synthase (Glyma.18G041100) gene after the *GmSHMT08* at the *Rhg4* locus. The sucrose synthase 1 (Glyma.15G182600) was reported frequently in SCN QTL mapping analysis, followed by the 6-phosphogluconate dehydrogenase (Glyma.08G254500) and the mitochondrial GmSHMT08m (Glyma.08G274400), showing the contribution of the glycolysis cycle to SCN resistance Appendix A.

Within the SCN QTLs, we were able to identify high frequency genes that belong to glycolysis (30 SCN QTLs), followed by cytoskeleton-related genes (18 SCN QTLs), glyoxylate cycle (13 SCN QTLs), redox hemeostasis (9 SCN QTLs), and ATP-mitochondrial-related genes (8 SCN QTLs) Appendix A.

### 2.4. Identification of Genes Related to Redox Homeostasis

The current study revealed many candidate genes related to redox hemostasis (Appendix A, Figure 3). The annotation of the set containing these genes showed the presence of four genes that encode glutathione S-transferase (GST), three genes encoding glutathione peroxidases, two genes encoding NAD(P)H dehydrogenase, and two genes encoding glutamate dehydrogenases Appendix A. The obtained mass spectrometry data are coherent with previous studies showing that modulation of the SHMT serine/glycine interconversion impact important maintenance of redox homeostasis occurs via both glutathione synthase and glutathione peroxidases [23]. RNAseq analysis showed that transcripts from the previous 11 genes were significantly induced under SCN infection in both root and leaves (Figure 2E, Appendix A).

### 2.5. Glycolysis Cycle in Response to SCN Infection

Glycolysis cycle provides several products that support nematodes growth. This study showed several candidate genes related to the glycolysis cycle including four sucrose synthase 1-related genes, three glyceraldehyde 3-phosphate dehydrogenase, three enolases, two glutamine synthetases, two 6-phosphogluconate dehydrogenases, 1 fructose-bisphosphate aldolase, and one glutamyl-tRNAGlu reductase Appendix A. Transcripts from all these 16 genes were induced under SCN infection in both root and leaves (Figure 2E, Appendix A).

### 2.6. Identification of Gene-Related Glyoxylate Cycle

Glyoxylate products support early nematode development. Mass spectrometry analysis revealed the presence of many candidate genes related to the glyoxylate cycle. The annotation of the set containing these genes showed the presence of five malate dehydrogenases (MDH1), five NADP-dependent malic enzymes, four glycine dehydrogenases (GLDC, gcvP), and two NADPH-specific isocitrate dehydrogenases Appendix A. Transcripts from all these 16 genes were induced under SCN infection in both root and leaves (Figure 2E, Appendix A).

### 2.7. Identification of Succinyl-CoA, Serine/Glycine, and Heme-Related Genes

Several genes related to the serine and glycine synthesis were identified by mass spectrometry including six serine hydroxymethyltransferases (GmSHMT08c, GmSHMT02m, GmSHMT08m, GmSHMT09m, GmSHMT14m, and GmSHMT18m), three glycine hydroxymethyltransferases, and one glycine decarboxylase. Additionally, mass spectrometry showed the presence of two dihydrolipoyllysine-residue succinyltransferases and two methylmalonate-semialdehyde dehydrogenases Appendix A. Dihydrolipoyllysine-residue succinyltransferase and Methylmalonate-semialdehyde dehydrogenase are key enzyme to synthesize Succinyl-CoA that—together with glycine, the SHMT product—produce the ALA, an important component of and precursor to the production of heme. It is well known that SCN requires heme source for its survival. RNAseq data showed that transcripts from all previous 12 enzymes were induced under SCN infections (Figure 2E, Appendix A).

### 2.8. Identification of Cytoskeleton-Related and ATP-Mitochondrial-Related Genes

Several components of cytoskeleton-related genes were found, including two actin, two actin-7, one tubulin-A, and nine tubulin beta-4 in addition to several ATP- and ADP-mitochondrial-related genes, such as two ADP/ATP carrier protein 1, an ATP synthase subunit beta-1, an ATP synthase subunit alpha, and two ADP-ribosylation factor 2-B-like genes that regulate the interaction of tubulin-folding cofactor D with native tubulin Appendix A. Interestingly, transcripts from all the 21 genes were induced under SCN infection in both root and leaves (Figure 2E, Appendix A).

### 2.9. In silico Analysis of the GmSHMT08 THF Cofactor Binding Sites and PLP Cofactor Binding and Catalysis Sites

Mass spectrometry analysis pointed to the importance of the GmSHMT08 in the interconversion of serine and glycine, two substrates that are essential for redox hemeostasis, serine/glycine conversion, glyoxylate cycle, glycolysis, succinyl-CoA, and heme biosynthesis (Figure 3). This interconversion relies on two essential sites at the GmSHMT08 enzyme, the PLP and the THF cofactor sites. To carry out the GmSHMT08 protein homology modeling, an available SHMT crystal structure from of serine hydroxymethyltransferase from glycine max cultivar Essex already complexed with PLP-glycine and 5-formyltetrahydrofolate residues was used as the template (Figure 4). Next, all four THF cofactor and six PLP cofactor sites at the GmSHMT08 were mapped against the model (Figure 4). To visualize the effect of the site directed mutagenesis on the THF and PLP residues, rotamers tools have been used to mutate the four THF and 6 PLP residues on the GmSHMT08 protein model (Figure 5). The PLP molecule binds to different residues in the GmSHMT08 PLP binding pocket, as shown in Figure 4. Lys-244 forms a covalent Schiff base linkage (internal aldimine) with PLP (Figure 4). Nearby residues from both chains of the obligate dimer (Tyr-59′, Glu-61′, Ser-107, Asp-215, Thr-241, Arg-250 (prime indicates chain B) (Figure 4)) assure conserved interactions with the phosphate, N1, and O3 hydroxyl of PLP, whereas the pyridine ring of PLP stacks against His-134. The SHMT8-PLP-Gly complex represents an intermediate step of the THF-dependent catalytic mechanism, in which L-Ser attacks the Schiff base linkage between Lys-244 (Figure 4) and PLP to form a PLP-Ser external aldimine. Formaldehyde is next liberated when the active site general base deprotonates the hydroxyl side chain of L-Ser. Once synthesized, formaldehyde is next attacked by THF N5, transferring the side chain of L-Ser to THF, resulting in an external aldimine/quinonoid product called PLP-Gly.

In silico analysis revealed that the mutated GmSHMT08^ΔL129A^, GmSHMT08^ΔL135A^, GmSHMT08^ΔF284A^, GmSHMT08^ΔN374A^, GmSHMT08^ΔY59A^, GmSHMT08^ΔG106A^, GmSHMT08_ΔG107A_, GmSHMT08_ΔH134A_, GmSHMT08_ΔS190A_, and GmSHMT08_ΔH218A_ alleles and the two polymorphisms between the SCN resistant cultivar Forrest and the SCN susceptible Essex are predicted to impact negatively their conserved interactions with the phosphate, N1, and O3 hydroxyl of PLP near the PLP binding pocket (Figure 5). These mutations are expected to affect the GmSHMT08’s ability to bind PLP substrate and the interconversion of serine/glycine and tetrahydrofolate (THF)/5,10-methylene THF, which may impact resistance to SCN.

### 2.10. Re-Analysis of the EMS-Induced GmSHMT08 Mutations Reveal Their Potential Impact on PLP/THF Cofactor Binding and Catalysis

To gain more insight into the impact of the eighteen EMS *Gmshmt08* mutants identified earlier [2,6,7] on the PLP/THF cofactor binding and catalysis, all previous mutants were mapped and mutated using the rotamers tool. E61K and G71D are found close to the Tyr59 residue, that is required for PLP cofactor catalysis. E61K, G62S, and P285S are located very close to the Phe284 that is required for THF binding. Disruption of this site profoundly altered substrate binding and catalytic activity in *E. coli* [24]. Therefore, *Gmshmt08_E61K_*, *Gmshmt08_G62S_*, and *Gmshmt08_p285S_* mutations are likely to have the same conformational deficiency by impacting the THF cofactor binding and PLP cofactor catalysis at the GmSHMT08.

The Forrest polymorphism R130P and *Gmshmt08_G132D_* mutant were located close to the Leu129 residue that is required for THF binding. *Gmshmt08_G132D_* is also close to the two essential and conserved histidine residues: His134 and His137. Since proline has a conformational rigidity due to its direct incorporation of the α-carbon into its side chain, this may cause drastic conformational changes, interfering with this catalysis (Figure 6).

Both polymorphic substitutions between Essex and Forrest (P130R and N358Y) were shown to impact the mobility of a loop near the entrance of the THF binding site at the GmSHMT08 protein, resulting in reduced affinity for folate substrate, subsequently impairing the enzymatic activity of GmSHMT08 [20]. *Gmshmt08_G357R_* mutation is located one residue away from the Forrest polymorphic substitution N358Y and therefore is predicted to impact the THF site’s binding to folate. Another mutation, *Gmshmt08_Q226*_*, which resulted in a loss of SCN resistance in Forrest, was mapped close to His218 residues involved in PLP cofactor binding.

*Gmshmt08_G106S_* is located at the PLP cofactor binding site Gly106, in addition to being located very close to the other PLP cofactor binding site Ser107. *Gmshmt08_G106S_*, *Gmshmt08_A302V_*, and *Gmshmt08_L299F_*, were closely located to the Thr-241 and Arg-250 where the pyridine ring of PLP stacks against His-134 (PLP catalysis). In addition, the previous mutations were mapped close to the Lys-244 that forms a covalent Schiff base linkage (internal aldimine) with PLP.

### 2.11. Functional Validation of the GmSHMT08 THF Cofactor Binding Sites and Their Role in SCN Resistance

The four THF cofactor binding sites (L129, L135, F284, N374) at the GmSHMT08 protein contribute to the interconversion of tetrahydrofolate (THF) and 5,10-methylene THF [7]. To test the effect of each THF binding sites on SCN resistance, we introduced independent mutations in each one of the four THF-related residues at the *GmSHMT08* coding sequence from the resistant Forrest allele, then overexpressed it in the *ExF12* RIL, carrying the SCN-resistant *GmSNAP18^+^* from Forrest and the SCN-susceptible *GmSHMT08^−^* allele from Essex. To conduct the GmSHMT08^ΔL129A^, GmSHMT08^ΔL135A^, GmSHMT08^ΔF284A^, GmSHMT08^ΔN374A^, and GmSHMT08 overexpression analysis, the 1416-bp nucleotide coding sequence of the different *GmSHMT08* alleles were overexpressed under the control of a soybean ubiquitin promoter using a transgenic hairy root system. Interestingly, unlike the *GmSHMT08* wild-type allele from Forrest that reduced the cyst numbers of the *ExF12* RILs by 91% in the susceptible *ExF12* background, reductions of the cyst numbers at the induced mutations, including GmSHMT08^ΔL129A^, GmSHMT08^ΔL135A^, GmSHMT08^ΔN374A^, and GmSHMT08^ΔF284A^, were limited to 50%, 58%, 66%, and 78%, respectively (Figure 7). Thus, induced mutations at GmSHMT08^ΔL129A^, GmSHMT08^ΔL135A^, GmSHMT08^ΔN374A^, and GmSHMT08^ΔF284A^ affected the GmSHMT08’s ability to reduce the number of cysts by 40%, 32%, 24%, and 12%, respectively. Statistical analysis showed that of the four THF sites, site directed mutagenesis of the GmSHMT08^ΔL129A^, GmSHMT08^ΔL135A^, and GmSHMT08^ΔF284A^ were significantly different (*p* < 0.0001) from the GmSHMT08 wild-type allele. The mutation at the GmSHMT08^ΔN374A^ THF residue presented the lowest reduction of cyst numbers.

### 2.12. Functional Validation of the GmSHMT08 PLP Cofactor Binding and Catalysis Sites Points to Their Involvement in SCN Resistance

Four PLP cofactor binding sites (G106, G107, S190A, H218) and two PLP cofactor catalysis sites (Y59 and H134) at the GmSHMT08 protein are involved in the interconversion of serine and glycine [7]. To test the real effect of each PLP cofactor binding and catalysis sites on SCN resistance, we introduced independent mutations in each one of the six PLP related residues at the *GmSHMT08* coding sequence from the resistant Forrest allele, then overexpressed it in the *ExF12* RIL. To conduct the GmSHMT08^ΔY59A^, GmSHMT08^ΔG106A,107A^, GmSHMT08^ΔH134A^, GmSHMT08^ΔS190A^, GmSHMT08^ΔH218A^ and GmSHMT08 overexpression analysis, the 1416-bp nucleotide coding sequences of the different *GmSHMT08* alleles were overexpressed under the control of a soybean ubiquitin promoter using a transgenic hairy root system. Surprisingly, unlike the *GmSHMT08* wild-type allele from Forrest that reduced the cyst numbers of the *ExF12* RILs by 91% in the SCN-susceptible *ExF12* background, reductions of the cyst number at the induced mutations, including GmSHMT08^ΔS190A^, GmSHMT08^ΔY59A^, GmSHMT08^ΔG106A,107A^, GmSHMT08^ΔH218A^, and GmSHMT08^ΔH134A^ were limited to 6%, 42%, 61%, 61%, and 63%, respectively (Figure 7). Thus, induced mutations at GmSHMT08^ΔS190A^, GmSHMT08^ΔY59A^, GmSHMT08^ΔG106A,107A^, GmSHMT08^ΔH218A^, and GmSHMT08^ΔH134A^ affected the GmSHMT08’s ability to reduce the number of cysts by more than 84%, 48%, 29%, 29%, and 27%, respectively. Most importantly, the PLP cofactor binding site (S190A) and the PLP cofactor catalysis site (Y59) presented higher impact on SCN resistance when compared to the four THF binding sites that were tested previously.

## 3. Discussion

Soybean cyst nematode is the most destructive pathogen to soybeans [3]. Most of the efforts to understand the SCN resistance mechanism were focused on deciphering the genes for resistance to SCN within two types of SCN resistance: the PI88788, which uses the *rhg-1b*; and Peking-type resistance, which uses a combination of *rhg1-a* and *Rhg4* loci [2,6]. Although the gene that confers resistance to SCN at the *Rhg4* locus has been cloned and identified a decade ago [2]; the involvement of the four THF cofactor binding sites, four PLP cofactor binding sites, two PLP cofactor catalysis sites at the GmSHMT08 in resistance to SCN has not been revealed yet.

The current study revealed the presence of enriched cycles that mainly use glycine as a substrate, including the glyoxylate cycle, redox homeostasis, and heme biosynthesis. Several key enzymes involved in the glycolysis cycle were also identified, which is coherent with QTL analysis studies that were reported earlier Appendix A). Although glucose and glutamine are the main sources that are used to maintain the glycolysis pathway, serine plays an essential role in the glycolysis pathway through de novo serine biosynthesis. Serine derived from a branch of glycolysis can be reintegrated into the glycolysis pathway to synthesize pyruvate but can also be converted to glycine, which provides carbon units for one carbon metabolism. One carbon metabolism is essential for the synthesis of proteins, lipids, nucleic acids, and other precursors through a complex metabolite network based on the chemical reactions of folate compounds. Thus, the main products of SHMT, serine, and glycine provide precursors for the biosynthesis of proteins, nucleic acids, and lipids, which are essential for both host and pathogen growth. The war between nematodes and soybean for metabolites and how plants can fight underground attacks is complex and still requires investigation [25].

Proteomic and transcriptomic assays of SCN-infected soybeans identified seven enzymes that play essential roles in the glycolysis and four enzymes in the glyoxylate cycle. Previous studies reported the ability of nematodes to steal nutrients from host plants [26]. Soluble sugars, such as fructose, glucose, and sucrose, were previously found to increase significantly in tomato leaves and roots during early infection by root-knot nematodes (RKNs) [27], which is coherent with the 7 genes identified at the glycolysis cycle in the current study. Another plant-parasitic nematode, *Heterodera schachtii*, has been shown to stimulate plant root cells to form syncytial feeding structures which synthesize all nutrients required for successful nematode development [27,28]. During glycolysis, a series of enzymatic reactions will convert sugars, typically sucrose to glucose, fructose, and then to pyruvate [29]. Products derived from glycolysis and glyoxylate cycle support early nematode development [30]. Nematodes will metabolize energy through the standard metabolic pathways, which is reflected by high metabolic activity, elevated sucrose levels, and the formation of starch [31,32]. The root cells affected by nematode attack show altered metabolisms—especially increased allocation of soluble sugars. Sugar importation into syncytia follows the symplasmic path during later stages of development [33,34,35].

On the other hand, it is known that glycine powers the biosynthesis of heme. Since SHMT catalyzes the conversion of serine to glycine, any disruption of the PLP/THF cofactor binding/catalysis sites, as shown in the EMS *GmSHMT08* mutants and by site-directed mutagenesis, may negatively impact the production of glycine and therefore the biosynthesis of heme. Heme is considered a major nutrient for nematodes from the plant host. Nematodes such as *Rhabditis maupasi*, *Caenorhabditis elegans*, and *Heterodera glycines* require heme source or any related iron porphyrin for feeding and survival [27,36,37]. This may explain the presence of four heme-related genes that were obtained via LC-MS and their differential expression during SCN infection. Two out of the four identified genes (Glyma.08G066600 and Glyma.07G183600, belonging to the malonate-semialdehyde dehydrogenase gene family), were mapped at two SCN QTLs in previous studies Appendix A.

Several genes belonging to the redox hemostasis pathway that were identified by LC-MS in the current study, such as glutathione peroxidases, NADP(H) dehydrogenase, glutathione S-transferases (GSTs), and glycine dehydrogenases, were differentially expressed under SCN infection. GSTs catalyze the conjugation of glutathione (GSH) to xenobiotic substrates for detoxification [38,39,40]. GST activity is dependent upon GSH supply from the glutathione synthetase enzyme and the activity of some transporters to remove GSH conjugates from the cell [41,42]. Most of the identified ROS proteins from LC-MS were differentially expressed under SCN infection. This is coherent with previous transcriptomic analysis, in which both glutathione peroxidase and glutathione transferase transcripts, among other ROS-scavenging enzymes, were shown to be significantly modulated under SCN infection (in syncytia) [43].

Recently, mitochondrial *OsSHMT* and *NbSHMT* have been demonstrated to play a role in broad-spectrum resistance via the ROS pathway [44]. Cytoskeletons (i.e., microtubules) play an important role during the intracellular transport of mitochondria [45,46]. The interaction of some mitochondrial components with certain cytoskeletal proteins was found to be involved in the coordination of mitochondrial function [47,48]. In fact, interaction between the microtubule-associated C4HC3-type E3 Ligase (MEL) and the mitochondrial SHMT1 leads to SHMT1-dependent mitochondrial ROS generation, activation of MAPK cascades, and reprogramming of defense-related transcripts, ultimately leading to attenuated pathogen invasion. The interacting MEL-SHMT1 complex mediates regulation of plant immunity involving microtubules and mitochondria. Infections by multiple pathogens induce MEL transcription. This is followed by the formation of MEL homodimers, which activate MEL E3 ligase activity, subsequently triggering SHMT1 degradation by the 26S [44]. Cytoskeleton-including actin filaments are dynamic structures that can grow and shrink rapidly via the addition or removal of tubulin proteins. During cellular homeostasis responses, mitochondria organelles are considered the major source for the generation of intracellular ROS by supplying ATP and biosynthetic intermediates for redox, cell death, and energy metabolism [49,50,51,52].

The current study found several potential substrates/components of cytoskeleton including Actins, Tubulin A, Tubulin beta-4, and several ATP and ADP mitochondrial related genes including ADP/ATP carrier protein 1, ATP synthase subunit beta-1, ATP synthase subunit alpha, and ADP-ribosylation factor 2-B-like that regulates the interaction of tubulin-folding cofactor D with native tubulin. Tubulin and actin cytoskeletons have been continuously reported to be implicated in plant defense against pathogenic fungi, oomycetes, and bacteria [52,53,54,55]. We also found the presence of cyclophilin, which are known to be modulated by microtubules [44]. The role of cyclophilin in plant pathogenesis has been reported earlier [56]. The cyclophilin GmCYP1 (Glyma.11G098700) has been suggested to play a role in soybean defense via its interaction with the isoflavonoid regulator GmMYB176 [57], which is known to play major roles in resistance to cyst nematodes in *Arabidopsis* [58] and in SCN [59]. Microtubule disruption of hematopoietic cells cause a dramatic subcellular redistribution of cyclophilin-A and pin1 from the nucleus to the cytosol and plasma membrane [60]. Another microtubule, MAP65-3 microtubule-associated protein, has been shown to be essential for cytokinesis in somatic cells and also play an important role during nematode-induced giant cell ontogenesis in *Arabidopsis* [61]. In fact, MAP65-3 is associated with mini cell plates that are required for the formation of a functional nematode feeding cell. In giant cell *map65-3* mutants, a defect in mini cell plate formation prevents the development of functional feeding cells, which resulted in the death of the nematode [61]. The identification of several cytoskeleton components from the current study reinforces their involvement in resistance to SCN, which is coherent with QTL SCN analysis where the identified 22 cytoskeleton-related and ATP mitochondrial-related genes were mapped to 26 reported SCN QTLs.

## 4. Material and Methods

### 4.1. Protein Extractions and Immunoprecipitation Using GmSHMT08 Antibodies

Forrest and Essex soybean cultivars were infected using SCN (HG0), as described earlier [62]. Root and leaf samples from three biological replicates containing five SCN (HG0)-infected and five non-SCN-infected soybeans were washed and frozen in liquid nitrogen three days after infection. Total root proteins from SCN-infected and non-infected soybean “Forrest” and “Essex” cultivars were extracted in a lysis buffer containing 5mM DTT, 1% (*v*/*v*) NP40, 1mM sodium molybdate, 1 mM NaF, 1 mM PMSF, 1.5 mM Na3VO4, 100 mM NaCl, 2 mM EDTA, 50 mM Tris–HCl at pH 7.5, 10% (*v*/*v*) glycerol, and one tablet from the plant protease and phosphatase inhibitors at 1:100 mL (Thermo Scientific), as previously shown [21]. Coomassie Bradford Protein Assay Kit was used to quantify protein concentrations (Thermo Fisher Scientific, Waltham, MA, USA). For in planta immunoprecipitation analysis, anti-GmSHMT08 polyclonal antibodies [21] were immobilized in a column (Pierce Co-Immunoprecipitation Kit). Then, immunoblot analysis of root protein fraction samples from soybean Forrest and Essex was incubated overnight with the immobilized antibodies. After three washes, the associated proteins were eluted as described by the Pierce Co-Immunoprecipitation Kit. The eluted fraction was then used for mass spectrometry analysis.

### 4.2. Mass Spectrometry Analysis

Peptide digestion, microsequencing analyses, and protein characterization of the SHMT-associated proteins from non-infected and SCN-infected Forrest and Essex roots 3 DAI were carried out in the Charles W Gehrke Proteomics Center at the University of Missouri-Columbia, as previously shown [21]. The eluted fractions obtained from the immunoprecipitation experiment using anti-GmSHMT08 polyclonal antibodies were briefly subjected to lyophilization. Then, all proteins were subsequently digested with trypsin, resulting in one main fraction representing the three biological replicates. Furthermore, samples were acidified, lyophilized, and re-suspended in 21 µL of a 5% acetonitrile, 0.1% formic acid solution, and peptides were analyzed via LC-MS (18 µL injection), as previously described [63]. Liquid chromatography gradient conditions were carried out as previously shown [63]. The Proxeon Easy nLC HPLC system was attached to an LTQ Orbitrap XL mass spectrometer. BSA was used for quality control on the column. Searches of Swiss-Prot-all species and NCBI-Gmax were conducted using Sorcerer-Sequest.

### 4.3. RNA-seq Library Preparation and Analysis

Four plant soybean tissues were used for RNA-seq, including SCN-infected (3 DAI) soybean root, non-SCN-infected soybean root, SCN-infected (3 DAI) soybean leaves, and non-SCN-infected soybean leaves. Three biological replicates that correspond to three independent experiments where each experiment contained five SCN (HG0) infected and five non-SCN-infected soybean plants were washed and frozen in liquid nitrogen three days after infection. Total RNA for each sample was extracted from 100 mg of frozen grounded samples using RNeasy QIAGEN KIT (Cat. No./ID: 74004, Germantown, Maryland). Total RNA was treated with DNase I (Invitrogen, Carlsbad, CA, USA). RNA-seq libraries preparation and sequencing were performed at Novogene INC. (Cambridge, UK) using Illumina NovaSeq 6000. The four libraries were multiplexed and sequenced in two different lanes generating 20 million raw pair end reads per sample (150 bp). Quality assessment of sequenced reads was performed using fastqc version 0.11.9 [64]. After removing the low-quality reads and adapters with trimmomatic version V0.39 [64], the remaining high-quality reads were mapped to the soybean reference genome Wm82.a2.v1 using STAR, version v2.7.9 [65,66]. Uniquely mapped reads were counted using Python package HTseq v0.13.5 [67]. Read count normalization and differential gene expression analysis were conducted using the Deseq2 package v1.30.1 [68] integrated in the OmicsBox platform from BioBam (Valencia, Spain). DEGs were considered significant if *p* value < 0.05, Log_2_FoldChange no less than ±0.6. Expression profiling was visualized through a heatmap using Heatmapper [22].

### 4.4. Cloning the Forrest GmSHMT08 WT and Site Directed Mutagenesis

The *GmSHMT08* coding sequence from the Forrest WT (*Rhg4*) was amplified from soybean Forrest root cDNA via RT-PCR using the GmSHMT08c-AscI-Fw primers (ggcgcgccATGGATCCAGTAAGCGTGTGGGGTA) and the GmSHMT08c-AvrII-Rv primers (ggatccCTAATCCTTGTACTTCATTTCAGATACC) and cloned into the *pG2RNAi2* vector under the control of the soybean ubiquitin (GmUbi) promoter [21,23]. Cloning was carried out between *AscI* and *AvrII* cloning sites at the *pG2RNAi2* vector to generate *pG2RNAi2::GmSHMT08*, which was used as a positive control Appendix A. All mutations described in this study were introduced using site directed mutagenesis. The following mutated residues, GmSHMT08^ΔL129A^, GmSHMT08^ΔL135A^, GmSHMT08^ΔF284A^, GmSHMT08^ΔN374A^, GmSHMT08^ΔY59A^, GmSHMT08^ΔG106A^, GmSHMT08^ΔG107A^, GmSHMT08^ΔH134A^, GmSHMT08^ΔS190A^, and GmSHMT08^ΔH218A^, were cloned into the pG2RNAi2 vector to generate *pG2RNAi2::GmSHMT08^ΔL129A^*, *pG2RNAi2::GmSHMT08^ΔL135A^*, *pG2RNAi2::GmSHMT08^ΔF284A^*, *pG2RNAi2::GmSHMT08^ΔN374A^*, *pG2RNAi2::GmSHMT08^ΔY59A^*, *pG2RNAi2::GmSHMT08^ΔG106A^*, *pG2RNAi2::GmSHMT08^ΔG107A^*, *pG2RNAi2::GmSHMT08^ΔH134A^*, *pG2RNAi2::GmSHMT08^ΔS190A^*, and *pG2RNAi2::GmSHMT08^ΔH218A^* constructs, respectively Appendix A. All clones were target-sequenced to confirm that the genes and associated mutations were inserted correctly Appendix A.

### 4.5. Genotyping of ExF RIL Population

The ExF12 RIL used for composite hairy root soybean transformation carrying the resistant *GmSNAP18^+^* allele from Forrest but the susceptible *GmSHMT08^−^* allele from Essex was developed and genotyped as described by [62].

### 4.6. Transgenic Soybean Composite Hairy Root

The functional characterization of the four THF binding sites (L129, L135, F284, N374), four PLP binding sites (G106, G107, S190A, H218), and the two PLP catalysis sites (Y59 and H134) at the GmSHMT08 protein has been validated using the transgenic hairy root system experiment. Williams 82 composite hairy roots transformed with pG2RNAi2:: empty vectors were used as a negative control. The pG2RNAi2 vector has a sGFP-selectable marker in planta [21,23]. Transgenic Williams 82 composite hairy roots transformed with *pG2RNAi2::GmSHMT08* and the ten different mutated GmSHMT08 PLP and THF cofactor binding/catalysis sites (*pG2RNAi2::GmSHMT*^ΔPLP^ and *pG2RNAi2::GmSHMT*^ΔTHF^) were produced by injecting agrobacterium bacterial suspensions three times into the hypocotyl directly below soybean cotyledons using a 3 mL needle (BD#309578) as shown earlier [21]. After injection, composite hairy roots from at least 50 independent soybean transgenic plants per construct were grown and propagated in medium vermiculite. Transgenic soybeans were covered with plastic humidity domes sprayed consistently with water, maintained in a growth chamber for 1–2 weeks, and fertilized once per week with NPK 20-20-20 fertilizer. GFP-positive composite hairy roots at ~2–3 inches long were transferred into a steam-pasteurized sandy soil and packed into plastic containers as mentioned earlier [62]. Each container held 25 tubes and was suspended over water baths maintained at 27 °C. At least 15 plants from the control lines (WI82 and ExF12) were arranged in a randomized complete block design. Two days after transplanting, each plant was inoculated with ~2000 SCN (HG0) eggs. After 30 days, cysts were counted under a stereomicroscope. The experiment was independently conducted three times to obtain a minimum of 15 to 20 independent composite hairy root lines per construct per experiment. The results were plotted and analyzed for statistical significance by using analysis of variance (ANOVA) using the JMP Pro V12 software as described earlier.

### 4.7. GmSHMT08 TILLING Mutants

The availability of new crystal structure of the GmSHMT08 in soybeans [20] enhanced our knowledge of how the previously identified 18 EMS *Gmshmt08* TILLING mutants can affect the PLP/THF cofactor binding and catalysis sites. Thus, we performed in silico mutational analysis of the thirteen *Gmshmt08* EMS mutants that were identified using forward genetic screening [6], the three *Gmshmt08* EMS mutants identified using forward genetics [7], and the two *Gmshmt08* EMS mutants that were identified using Gel-TILLING [2].

### 4.8. Modeling of GmSHMT08 Protein, PLP and THF Cofactor Sites

Homology modeling of a putative GmSHMT08 protein structure was conducted using Deepview and Swiss-Model Workspace software, as previously shown [21,23]. Briefly, protein sequences from Forrest and the available SHMT crystal structure from soybean Glycine max cultivar Essex (PDB accession 6uxj.1) were used as templates. Residues 2–470 were modelled against their corresponding template with a sequence identity of 99.57% (according to the Protein Data Bank database). The structure of serine hydroxymethyltransferase from Glycine max cultivar Essex was complexed with PLP-glycine and 5-formyltetrahydrofolate residues [20]. Visualization of the THF cofactor binding sites (L129, L135, F284, N374), PLP cofactor binding and catalysis residues (Y59, G106, G107, H134, S190A, H218)—in addition to the two polymorphisms (P130R and N358Y)—and the 18 EMS-induced GmSHMT08 mutations was performed using the UCSF Chimera package [69]. To study the impact on the THF/PLP cofactor binding/catalysis and EMS mutations that were located close to the PLP/THF cofactor sites, the mapped induced mutations at the PLP cofactor sites, THF cofactor sites, and EMS-induced mutations were mutated using the structural editing tool from the UCSF Chimera package. Then, the rotamers tool that is incorporated within the Chimera package software was used to mutate the corresponding residues [24]. The rotamers tool allows amino acid sidechain rotamers to be viewed, evaluated, and incorporated into structures in which a given residue can be changed into different amino acids to predict the impact and effect of the mutations on the adjacent residues surrounding the mutated residue.

## 5. Conclusions

Our data are coherent with previous studies showing that glutathione peroxidase transcription, among other ROS scavenging enzymes, was significantly modulated under SCN infection in syncytia [51]. The *Arabidopsis thaliana Atshmt1-1* mutant showed a greater accumulation of H_2_O_2_, which is known to induce salicylic acid biosynthesis [70,71]. The implications of phytohormones, such as SA and CK, have been previously shown to be involved in a crosstalk between SCN-resistant genes (*GmSHMT08* and *GmSNAP18*) and SCN defense genes (*GmPR08-Bet VI*) [21]. Maintenance of a certain level of ROS homeostasis at low levels is required for parasitic nematodes to cause and maintain pathogenic disease [72,73]. However, disruption of this homeostasis (overaccumulation of ROS) can cause termination of syncytial formation or syncytial apoptosis [72,73,74]. Taken together, modulation of the SHMT serine/glycine interconversion may impact important maintenance of redox homeostasis that occurs via ROS. Maintenance of balanced SHMT expression appears to be highly important in plants. The current study uncovered for the first time the involvement of the interconversion reaction carried out by the serine hydroxymethyltransferase protein involving the two cofactors at the GmSHMT08c protein, the four THF cofactor biding sites, and the six PLP cofactor binding/catalysis sites in resistance to SCN.

## Figures and Tables

**Figure 1 ijms-23-11278-f001:**
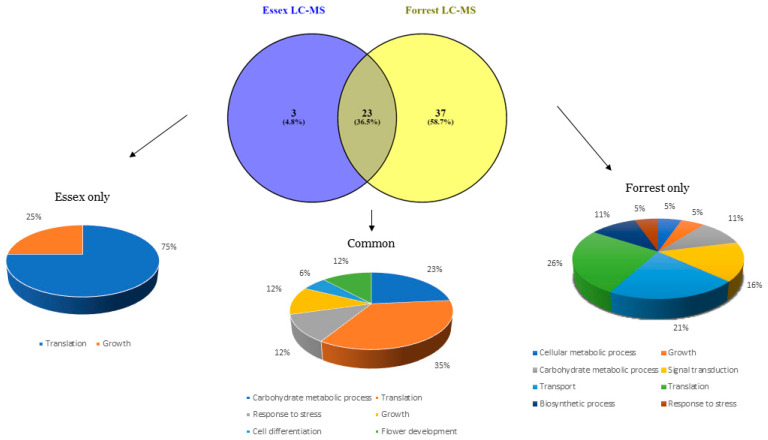
Venn diagram showing relationships between different biological processes obtained using mass spectrometry from Forrest and Essex roots under SCN infection (3 DAI). Thirty-seven proteins were found in the resistant genotype “Forrest” only and correspond to 8 different biological processes, while 3 proteins were identified in the susceptible genotype “Essex” and correspond to two biological processes. Twenty-three proteins were common between Forrest and Essex and correspond to six different biological processes.

**Figure 2 ijms-23-11278-f002:**
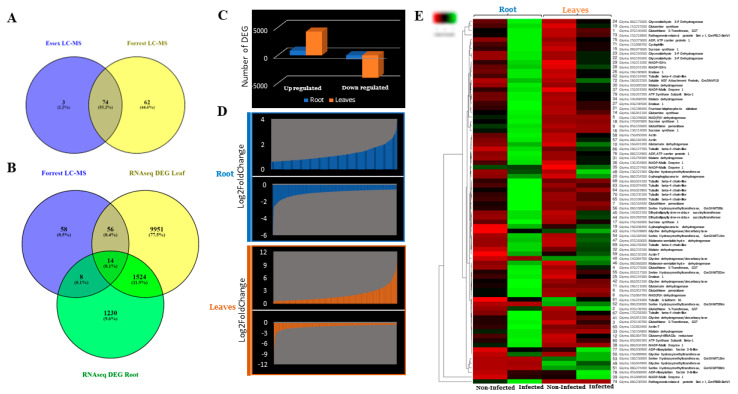
(**A**) Venn diagram showing relationships among different candidate genes obtained via mass spectrometry from Forrest and Essex. Sixty-two genes were identified in Forrest only, and 3 genes were identified in Essex only. The other 74 genes were common between Essex and Forrest. (**B**) Of 136 genes identified in Forrest MS, 78 were differentially expressed under SCN infection in the resistant Forrest roots and leaves. Of these 78 genes, 26 were differentially expressed in Forrest leaves, 8 were differentially expressed in Forrest roots, and 14 were differentially expressed in both Forrest roots and leaves. (**C**,**D**) Significantly up- and down-regulated genes by plant tissue. DEGs considered significant if *p* value < 0.05, log2foldchange no less than ±0.6. (**E**) Heatmap of all DEGs by log2FC (fold change) of induced response to SCN treatment of roots and leaves in Forrest. The expressions profiling was visualized through heatmap using Heatmapper [22].

**Figure 3 ijms-23-11278-f003:**
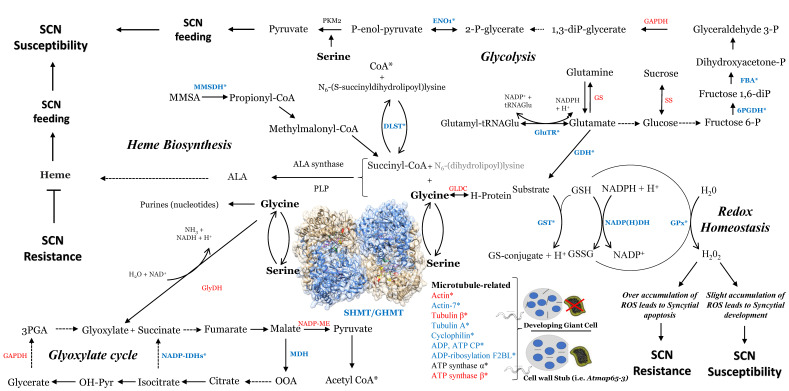
Summary of all proteins identified by mass spectrometry, revealing key components related to redox hemeostasis, serine/glycine conversion, glyoxylate cycle, glycolysis, succinyl-CoA, and heme-biosynthesis-related enzymes. All identified proteins belong mainly to processes and cycles that use glycine/serine as a substrate/precursor and are therefore directly related to the interconversion reaction carried out by the serine hydroxymethyltransferase protein. Enolase (ENO1), glyceraldehyde 3-P dehydrogenase (GAPDH), fructose-bisphosphate aldolase (FBA), 6-phosphogluconate dehydrogenase (6PGDH), sucrose synthase (SS), glutamine synthase (GS), glutamyl-tRNAGlu reductase (GluTR), glutamate dehydrogenase (GDH), glycine decarboxylase (GLDC), dihydrolipoyllysine-residue succinyltransferase (DLST), methylmalonate-semialdehyde dehydrogenase (MMSDH), glutathione S-transferase (GST), NADP(H) dehydrogenase (NADP(H)DH), glutathione peroxidase (GPx), glycine dehydrogenase (GlyDH), serine/glycine hydroxymethyltransferase (SHMT), NADP-malic enzyme (NADP-ME), NADPH-specific isocitrate dehydrogenase (NADP-IDHs), malate dehydrogenase (MDH), carrier protein (CP), factor 2-B-like (F2BL). Blue are enzymes/genes that were identified in Forrest only. Red genes are present in both Essex and Forrest. * Genes identified within or very close to QTL for SCN resistance.

**Figure 4 ijms-23-11278-f004:**
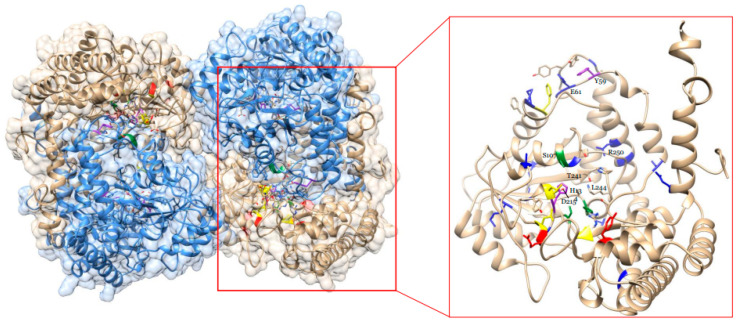
Homology modeling of an available SHMT crystal tetramer structure of a serine hydroxymethyltransferase from glycine max cultivar Essex complexed with PLP-glycine and 5-formyltetrahydrofolate residues used as template. On the right, on SHMT08 homomer showing all four mapped THF binding sites (yellow), four PLP binding sites (Green), two PLP catalysis sites (purple), two polymorphisms (red), and *Gmshmt08* EMS mutants (Blue).

**Figure 5 ijms-23-11278-f005:**
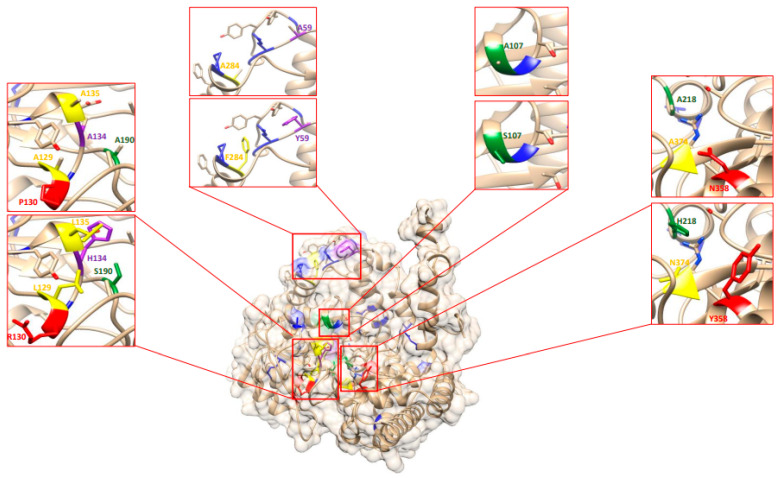
Homology modeling of one GmSHMT08 homomer showing mutated PLP and THF residues. Rotamers tools have been used to mutate the four THF and six PLP residues on the GmSHMT08 protein model to visualize the effect of the site directed mutagenesis on the THF and PLP residues. Four THF binding sites (yellow), four PLP binding sites (Green), two PLP catalysis sites (purple), two polymorphisms (red), and *Gmshmt08* EMS mutants (Blue).

**Figure 6 ijms-23-11278-f006:**
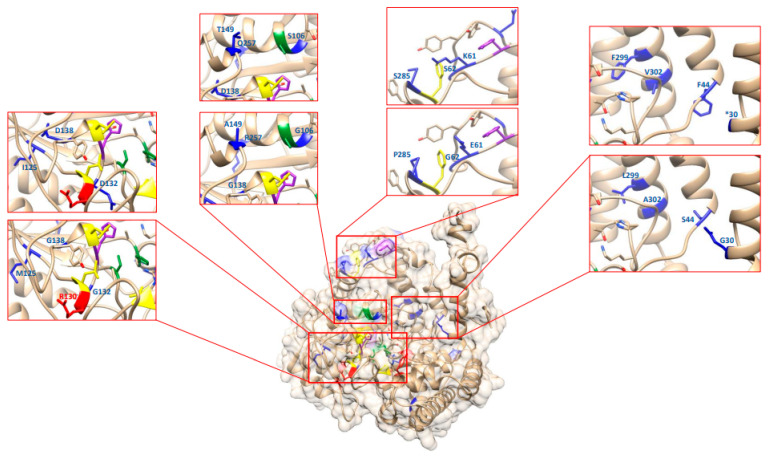
Homology modeling of one GmSHMT08 homomer, mapping EMS mutations at the GmSHMT08 protein. Rotamers tools have been used to mutate the 18 GmSHMT08 mutations. THF and PLP residues are shown. Four THF binding sites (yellow), four PLP binding sites (Green), two PLP catalysis sites (purple), two polymorphisms (red), and *Gmshmt08* EMS mutants (Blue).

**Figure 7 ijms-23-11278-f007:**
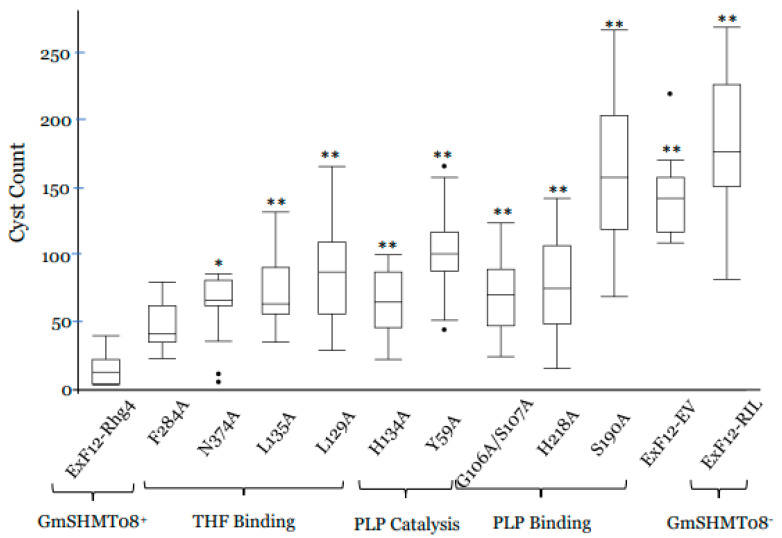
Composite hairy root transformation, overexpression analysis of the GmSHMT08^ΔPLP^ and GmSHMT08^ΔTHF^. Overexpression analysis in transgenic composite roots (ExF12 RIL GmSNAP18^+^/GmSHMT08^−^ transformed by pG2RNAi2::GmSHMT08^ΔL129A^, pG2RNAi2::GmSHMT08^ΔL135A^, pG2RNAi2::GmSHMT08^ΔF284A^, pG2RNAi2::GmSHMT08^ΔN374A^, pG2RNAi2::GmSHMT08^ΔY59A^, pG2RNAi2::GmSHMT08^ΔG106A^, pG2RNAi2::GmSHMT08^ΔG107A^, pG2RNAi2::GmSHMT08^ΔH134A^, pG2RNAi2::GmSHMT08^ΔS190A^, and pG2RNAi2::GmSHMT08^ΔH218A^. The experiments were repeated three times, and similar results were obtained. The data shown represent the averages and SD from all three biological repeats (n > 15). Asterisks and connecting letters indicate significant differences between the tested lines and the pG2RNAi2::GmSHMT08 (used as positive control) as determined by ANOVA (** *p* < 0.0001, * *p* < 0.01).

## Data Availability

Data about GmSHMT08 EMS mutations were deposited at the NCBI.

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
