# Peer review of "Proteomic, Transcriptomic, Mutational, and Functional Assays Reveal the Involvement of Both THF and PLP Sites at the GmSHMT08 in Resistance to Soybean Cyst Nematode"

_ijms, 2022, doi:10.3390/ijms231911278_

Round 1

Reviewer 1 Report

The manuscript aimed to elucidate the role of PLP and THF sites in soybean cyst nematode resistance. It is an important and exciting study, and the generated data might help to understand soybean cyst nematode resistance mechanisms. Overall, the work appears to have been performed in an acceptable manner, and the manuscript is well-written and structured. The methodologies used in this manuscript are also very relevant to achieving the study objectives.

Minor concerns:

L16: Please provide the full name for SCN when first time mentioned.

L98: Although the authors explained why using leaves for this in the result section, it might be good to discuss it here briefly.

L110: It might be a little unclear on sample material and treatments. How was the SCN infection being carried out? Perhaps, the authors should consider describing plant cultivars used, how the authors infected soybeans, etc. in the first section. If that was from the previous, please provide references.

L135: re-suspended in 21 ul of?

L140: "Four plant soybean tissues.....Three biological replicates containing five SCN infected...". This sentence might be unclear. What are tissues here referring to? Roots and leaves? Do three biological replicates mean the experiment was repeated three times and five plants each time?

L160: primer sequence used?

L316-320: Figure 2A was mentioned in L262, followed by Figures 3, 4, and 5. But Figures 2C and 2D appeared only in L316 and L320, respectively. It might be a good idea to follow the sequence so that the readers do not need to refer back to the figure.

L509-537: In my opinion, comparing human serine hydroxymethyltransferase with plants is okay; at least we can know their similarity or differences and probably further explore their potential applications. But would that become overclaim if having such a lengthy discussion on this? 

L609: Capital "C".

L640: Perhaps this could be split as a new section, i.e., conclusion?

The manuscript does not follow the format required by the journal, i.e., reference format (number in the text). Perhaps, the authors could recheck the guidelines. 

Author Response

Minor concerns:

L16: Please provide the full name for SCN when first time mentioned.

The full name of SCN was provided.

L98: Although the authors explained why using leaves for this in the result section, it might be good to discuss it here briefly.

This section has been updated as suggested by the reviewer Although soybean cyst nematodes infect soybean roots, leaves play important role by supplementing nematodes with most of the nutrients that they use to grow and complete their life cycle. In the current study, root and leave transcriptomic analysis of differentially expressed genes under SCN infection supported the data from LC-MS”

L110: It might be a little unclear on sample material and treatments. How was the SCN infection being carried out? Perhaps, the authors should consider describing plant cultivars used, how the authors infected soybeans, etc. in the first section. If that was from the previous, please provide references.

This section has been updated as suggested by the reviewer Forrest and Essex soybean cultivars were infected using SCN (HG0) as described earlier (Lakhssassi et al., 2017)

L135: re-suspended in 21 ul of?

This section has been updated samples were acidified, lyophilized, and re-suspended in 21 µL of a 5% acetonitrile, 0.1% formic acid solution, and peptides were analyzed by LC-MS (18 µL injection) as described earlier (Moraes et al., 2020). Liquid chromatography gradient conditions were carried out as shown earlier (Moraes et al., 2020). The Proxeon Easy nLC HPLC system is attached to an LTQ Orbitrap XL mass spectrometer”

L140: "Four plant soybean tissues.....Three biological replicates containing five SCN infected...". This sentence might be unclear. What are tissues here referring to? Roots and leaves? Do three biological replicates mean the experiment was repeated three times and five plants each time?

Yes. This paragraph has been rewritten: Four plant soybean tissues were used for RNA-seq including a SCN infected (3 DAI) soybean root, a SCN non-infected soybean root, a SCN infected (3 DAI) soybean leaves, and a SCN non-infected soybean leaves. Three biological replicates that correspond to three independent experiments where each experiment contained five SCN (HG0) infected and five SCN non-infected soybean plants were washed and frozen in liquid nitrogen at three days after infection.”

L160: primer sequence used?

The primer sequence used for cloning the GmSHMT08 CDS has been added.

L316-320: Figure 2A was mentioned in L262, followed by Figures 3, 4, and 5. But Figures 2C and 2D appeared only in L316 and L320, respectively. It might be a good idea to follow the sequence so that the readers do not need to refer back to the figure.

This section has been updated as suggested by the reviewer.

L509-537: In my opinion, comparing human serine hydroxymethyltransferase with plants is okay; at least we can know their similarity or differences and probably further explore their potential applications. But would that become overclaim if having such a lengthy discussion on this? 

This section has been removed as suggested by the reviewer.

L609: Capital "C".

This has been corrected.

L640: Perhaps this could be split as a new section, i.e., conclusion?

The last section has been split as a new section (Conclusion).

The manuscript does not follow the format required by the journal, i.e., reference format (number in the text). Perhaps, the authors could recheck the guidelines. 

Reference format has been updated to the IJMS style/format.

Reviewer 2 Report

The manuscript " Proteomic, transcriptomic, mutational and functional assays reveal the involvement of both THF and PLP sites at the GmSHMT08 in resistance to soybean cyst nematode" by Lakhssassi et al. used omics tools and functional assays to identify some genes and QTL responsible for resistance to soybean cyst nematode. The data are original and novel. This paper could be published with some clarifications as requested below.

 1- The English need to be improved in the whole manuscript.

2- The introduction needs to be more condensed.

3- In Materials and Methods, the details of MS-LC analysis and data acquisition (e.g., HPLC settings, etc.) are missing. Please add these details to the method section.

4- Results section is quite long. You could perhaps consider whether this could be further condensed. For example, Line 388-390 can be removed, as it is not the result.

5- The discussion section is quite long too. Many introductory materials in the discussion can be removed from the paper. Or for example, from line 485 to line 537, the authors discussed the presence and different roles of SHMT in animals, E. coli, and humans. Are these necessary to be discussed here? I suggest the author remove all these, discuss their results, and compare them with a similar study on other plants.

 More specific comments are included below:

Please present the full name at first use and use the abbreviation afterward. Examples are presented below:

Line 16. What is SCN?

Line 17.  What is PLP?

Line 29.  …. this finding may have relevance to designing SHMT inhibitors as a new chemotherapeutic in humans to fight against lung cancer cells, cardiovascular diseases, and anti-malarial agents. This is a considerable claim without evidence!!!

Line 37.

United States amounted to 46.06 billion U.S. dollars in 2020 (Shahbandeh, 2021). Soybean production, however, is affected by the presence of a microscopic parasitic roundworm, Soybean Cyst Nematode (SCN), which contributes dramatically to a 30% yield loss in soybean crops nationwide, causing an estimated 1.5 billion dollars in damage

These numbers don’t match. If there are 30% losses of $46.06 billion, the losses should be ~ $14 billion, not $1.5 billion!!!! In fact, $1.5 billion is 3% (not 30%) of $46.06 billion.

Author Response

In Materials and Methods, the details of MS-LC analysis and data acquisition (e.g., HPLC settings, etc.) are missing. Please add these details to the method section.

This section has been updated as requested by the reviewer samples were acidified, lyophilized, and re-suspended in 21 µL of a 5% acetonitrile, 0.1% formic acid solution, and peptides were analyzed by LC-MS (18 µL injection) as described earlier (Moraes et al., 2020). Liquid chromatography gradient conditions were carried out as shown earlier (Moraes et al., 2020). The Proxeon Easy nLC HPLC system is attached to an LTQ Orbitrap XL mass spectrometer.”

Results section is quite long. You could perhaps consider whether this could be further condensed. For example, Line 388-390 can be removed, as it is not the result.

The paragraph has been removed as requested by the reviewer.

The discussion section is quite long too. Many introductory materials in the discussion can be removed from the paper. Or for example, from line 485 to line 537, the authors discussed the presence and different roles of SHMT in animals, E. coli, and humans. Are these necessary to be discussed here? I suggest the author remove all these, discuss their results, and compare them with a similar study on other plants.

This section has been removed as suggested by the reviewer.

More specific comments are included below:

Please present the full name at first use and use the abbreviation afterward. Examples are presented below:

Line 16. What is SCN? Soybean cyst nematode was introduced on the text.

Line 17.  What is PLP? Pyridoxal phosphate has been introduced on the text.

Line 29.  …. this finding may have relevance to designing SHMT inhibitors as a new chemotherapeutic in humans to fight against lung cancer cells, cardiovascular diseases, and anti-malarial agents. This is a considerable claim without evidence!!!

This claim has been removed.

Line 37.

United States amounted to 46.06 billion U.S. dollars in 2020 (Shahbandeh, 2021). Soybean production, however, is affected by the presence of a microscopic parasitic roundworm, Soybean Cyst Nematode (SCN), which contributes dramatically to a 30% yield loss in soybean crops nationwide, causing an estimated 1.5 billion dollars in damage

These numbers don’t match. If there are 30% losses of $46.06 billion, the losses should be ~ $14 billion, not $1.5 billion!!!! In fact, $1.5 billion is 3% (not 30%) of $46.06 billion.

It has been corrected.

The authors would like to thank the reviewer for his suggestions and recommendations to improve the manuscript.